# Isotretinoin and Thalidomide Down-Regulate *c-MYC* Gene Expression and Modify Proteins Associated with Cancer in Hepatic Cells

**DOI:** 10.3390/molecules26195742

**Published:** 2021-09-22

**Authors:** Patricia Nefertari Ramírez-Flores, Paulina J. Barraza-Reyna, Alain Aguirre-Vázquez, María E. Camacho-Moll, Carlos Enrique Guerrero-Beltrán, Diana Resendez-Pérez, Vianey González-Villasana, Jesús Norberto Garza-González, Beatriz Silva-Ramírez, Fabiola Castorena-Torres, Mario Bermúdez de León

**Affiliations:** 1Centro de Investigación Biomédica del Noreste, Departamento de Biología Molecular, Instituto Mexicano del Seguro Social, Monterrey 64720, Mexico; nefertari.rmz.flws@gmail.com (P.N.R.-F.); paulinajaneth@hotmail.com (P.J.B.-R.); aguirre.alain1@gmail.com (A.A.-V.); maria.camachomo@imss.gob.mx (M.E.C.-M.); 2Departamento de Biología Celular y Genética, Facultad de Ciencias Biológicas, Universidad Autónoma de Nuevo León, San Nicolás de los Garza 66451, Mexico; diaresendez@gmail.com (D.R.-P.); vianeygonzalez_villasana@hotmail.com (V.G.-V.); 3Tecnológico de Monterrey, Escuela de Medicina y Ciencias de la Salud, Monterrey 64710, Mexico; enriqueguerrero@tec.mx (C.E.G.-B.); fcastorena@tec.mx (F.C.-T.); 4Departmento de Ciencias Básicas, Vicerrectoría en Ciencias de la Salud, Universidad de Monterrey, San Pedro Garza García 66238, Mexico; jesusnorbertogarza@hotmail.com; 5Centro de Investigación Biomédica del Noreste, Departamento de Inmunogenética, Instituto Mexicano del Seguro Social, Monterrey 64720, Mexico; silbear2002@yahoo.es

**Keywords:** Hepatocellular carcinoma, isotretinoin, thalidomide, *c-MYC*, HepG2 cells

## Abstract

Hepatocellular carcinoma (HCC) is the most common form of liver cancer. The number of cases is increasing and the trend for the next few years is not encouraging. HCC is usually detected in the advanced stages of the disease, and pharmacological therapies are not entirely effective. For this reason, it is necessary to search for new therapeutic options. The objective of this work was to evaluate the effect of the drugs isotretinoin and thalidomide on *c*-MYC expression and cancer-related proteins in an HCC cellular model. The expression of *c*-MYC was measured using RT-qPCR and western blot assays. In addition, luciferase activity assays were performed for the *c-MYC* promoters P1 and P2 using recombinant plasmids. Dose-response-time analyses were performed for isotretinoin or thalidomide in cells transfected with the *c-MYC* promoters. Finally, a proteome profile analysis of cells exposed to these two drugs was performed and the results were validated by western blot. We demonstrated that in HepG2 cells, isotretinoin and thalidomide reduced *c-MYC* mRNA expression levels, but this decrease in expression was linked to the regulation of P1 and P1-P2 *c-MYC* promoter activity in isotretinoin only. Thalidomide did not exert any effect on *c-MYC* promoters. Also, isotretinoin and thalidomide were capable of inducing and repressing proteins associated with cancer. In conclusion, isotretinoin and thalidomide down-regulate *c-MYC* mRNA expression and this is partially due to P1 or P2 promoter activity, suggesting that these drugs could be promising options for modulating the expression of oncogenes and tumor suppressor genes in HCC.

## 1. Introduction

Hepatocellular carcinoma (HCC) is the most frequent kind of liver cancer [1,2]. The common causes of HCC involve biotic and abiotic factors, such as hepatitis B and C virus infection, aflatoxin B1 ingestion, and cirrhosis due to alcohol, among others [3,4]. Due to the lack of symptoms during the early stages of HCC, the detection of the disease occurs in the advanced stages [5]. Patients with HCC have a prognosis of fewer than six months of survival if not treated [6]. HCC treatment includes surgical excision, partial or total transplantation, hepatic artery embolization, electrocautery, radiotherapy, and chemotherapy [5]. The eligibility criteria for these options depends on the stage of the development of cancer. For chemotherapy, sorafenib is a drug that extends survival for an average of 2.8 months; however, controversial results have been reported regarding its use [7]. Most patients treated do not survive the rest relapse and develop metastasis after five to seven years. To this view, many approaches have been suggested to include other molecules in the treatment of this kind of cancer [8,9,10,11,12,13]. Cancer origin and progression are orchestrated by the disruption of the balance of tumor suppressor genes and oncogenes. One of the oncogenes that has been widely studied in a plethora of cancers is *c-MYC*. The *c-MYC* gene encodes a transcription factor that regulates the expression of other genes involved in cell proliferation, growth, differentiation, and apoptosis [14,15]. It has been observed that overexpression of *c-MYC* is a frequent abnormality in human HCC; therefore, it has been considered to be a therapeutic target of great importance in other types of cancer [16]. The overexpression of *c-MYC* has been attributed to amplifications and epigenetic modifications such as hypomethylation of its promoter region [17]. The *c-MYC* gene has four promoters: P0, P1, P2 and P3. Promoters P0 and P3 do not contain a TATA box and they transcribe less than 5% of *c-MYC* mRNA [18]. Therefore, P1 and P2 transcribe most of the mRNA, 10–25% and 75–90%, respectively.

In this context, we have identified the need to search for new therapeutic adjuvants for the treatment of HCC, where *c-MYC* expression may be down-regulated. On this basis, drug repurposing or drug repositioning aims to search for new or secondary therapeutic targets of approved drugs, with the advantage that they have already been tested in clinical trials [19]. Isotretinoin (13-*cis*-retinoic acid), a drug primarily used to treat severe acne, has been repurposed as a trypanocidal drug [20] and for the treatment of neuroblastoma [21]. Likewise, thalidomide, a drug originally prescribed for morning sickness symptoms and leprosy, is another example of drug repurposing in some types of cancer [22]. This study aimed to investigate the effect of isotretinoin and thalidomide on *c-MYC* gene expression and other proteins associated with cancer in a human hepatic cell line, in order to assess their use alone or in combination with other HCC-specific drugs, such as sorafenib, as a promising option for modulation of the expression of oncogenes and tumor suppressor genes for the treatment of HCC.

## 2. Results

To evaluate the effect of isotretinoin and thalidomide on HepG2 cell viability, cells were exposed to the drugs at different concentrations for 24, 48, 72, and 96 h. Results revealed a significant reduction in cell viability (less than 30%) at 96 h for all doses of isotretinoin (*p* < 0.001) (Figure 1A). However, thalidomide did not have any effect on HepG2 viability (Figure 1B). For the following experiments, 10 µM isotretinoin and/or 10 µg/mL (38.72 mM) thalidomide were selected as sublethal working concentrations for 48 h exposures, for which no evident changes were observed. First, the effects of 10 µM isotretinoin or 10 µg/mL thalidomide on *c-MYC* expression in HepG2 cells were evaluated by RT-qPCR. The expression of *c-MYC* mRNA was significantly reduced by ~80% when cells were treated with isotretinoin or thalidomide (*p* < 0.05) (Figure 2A). However, this effect was not observed when cells were treated with the combination of both drugs; on the contrary, we observed a significant increase (*p* < 0.05) in *c-MYC* mRNA expression compared to control cells (Figure 2A). We then evaluated c-MYC protein expression by western blot analysis, by triplicate (Figure 2B), after 48 h of treatment with isotretinoin and thalidomide individually, but no significant effect on protein levels was observed (*p* > 0.05) (Figure 2C). This demonstrates that 10 µM isotretinoin or 10 µg/mL thalidomide affect *c-MYC* mRNA expression when used individually, but this is not observed at the protein levels. 

To determine the transcriptional activity of c-MYC promoters, plasmids bearing P1 (pGL3-Myc-P1), P2 (pGL3-Myc-P2), and the combination of P1 and P2 promoters (pGL3-Myc-P1-P2) fused to the luciferase reporter gene were transfected into HepG2 cells. The pGL3 Control Vector and pGL3 Basic Vector were included in the analysis as positive and negative controls, respectively. Luciferase activity was determined in cell lysates after 48 h and a significant increase in luciferase activity was observed between HepG2 cells transfected with pGL3-Myc-P2 and pGL3-Myc-P1-P2, and the promoterless pGL3 Basic Vector (*p* < 0.05) (Figure 3). A significant, modest increase in luciferase activity was also observed in pGL3-Myc-P1 compared to pGL3 Basic Vector (*p* < 0.05) (Figure 3). As expected, pGL3 Control Vector showed the highest activity, more than 400 relative units of luminescence, when compared to pGL3 Basic Vector (Figure 3). Our data confirm that the elements of the P1-P2 and P2 promoter region mainly regulate the transcriptional activity of the *c-MYC* gene.

Then, to evaluate whether isotretinoin and thalidomide regulate the transcriptional activity of the P2 and P1-P2 promoters, which control the highest activity of the *c-MYC* gene, HepG2 cells were transfected with pGL3-Myc-P2 and pGL3-Myc-P1-P2 constructs for the following experiments. After transfections with pGL3-Myc-P2 or pGL3-Myc-P1-P2 constructs, cells were treated with 10 µM isotretinoin or 10 µg/mL thalidomide, and luciferase activity was measured in the cell lysates. Treatment with 10 µM isotretinoin for 48 h significantly reduced (*p* < 0.05) luciferase activity in P2 and P1-P2 transfected cells by 11.3% (Figure 4A) and 7% (Figure 4B), respectively. However, there was no significant difference in the luciferase activity of P2 and P1-P2 promoters in transfected cells treated with thalidomide (Figure 4). 

To determine whether the down-regulation of c-MYC P2 promoter activity was time-dependent, pGL3-Myc-P2 transfected HepG2 cells were exposed to 10 µM isotretinoin for 24, 48, and 72 h. Our results show that the down-regulation of the P2 promoter activity was maintained over the 72 h of treatment with isotretinoin, with a significant difference compared to untreated transfected control cells (*p* < 0.05, Figure 5A). Likewise, the regulation of c-MYC P2 promoter activity was also dose-dependent, with a progressive reduction observed at 1 µM (*p* < 0.05), 5 µM (*p* < 0.05), and 10 µM (*p* < 0.05) isotretinoin after 24 h (Figure 5B). Interestingly, there was no effect on cells treated with 20 µM isotretinoin compared to control cells (Figure 5B). To summarize, the down-regulation of the c-MYC P2 promoter activity by the effect of isotretinoin was maintained even after 72 h of exposure to the drug. This effect was dose-dependent, with the highest repression observed at 10 µM (Figure 5B).

To identify other proteins related to cancer that could be modified by isotretinoin and thalidomide, a proteome profile array was used to detect differences in 84 cancer-related proteins in extracts from HepG2 cells exposed to 10 µM isotretinoin or 10 µg/mL thalidomide for 48 h. We observed two down-regulated and 13 up-regulated proteins in response to isotretinoin, and one down-regulated and 13 up-regulated proteins in response to thalidomide (Figure 6A). Interestingly, single treatments with isotretinoin or thalidomide had the same effect on some oncogenes and tumor suppressor genes. Proteins modified by the sole treatment with the drugs included the oncogenes AXL, CEACAM-5, SNAIL, and M-CSF; and the tumor suppressor proteins Decorin, DLL1 and p53 (Figure 6B). Interestingly, some proteins showed the opposite change due to the effect of the drugs. This includes the Dickkopf-1 protein, Dkk-1, which was down-regulated by isotretinoin but up-regulated by thalidomide.

To confirm the results of the proteome array, we evaluated the protein expression levels of M-CSF, SNAIL, and p53 by western blot (Figure 7A). The analysis of M-CSF (Figure 7B) and SNAIL (Figure 7C) did not show significant changes due to the effect of the drugs. This was not correlated to the change observed in the proteome array. A flagship protein in cancer, p53, also showed no significant changes due to the effect of the drugs (Figure 7D).

## 3. Discussion

The regulation of the c-*MYC* oncogene is an attractive target for the search for new or repurposed drugs due to its relevance in HCC development. For the first time, we have demonstrated that isotretinoin and thalidomide down-regulate *c-MYC* expression in HepG2 cells, and this reduction is due partially to *c-MYC* P1 and P2 promoter activity. Although we observed differences at the mRNA level, no difference was observed at the protein level, which may be due to the half-life of each molecule or the molecular mechanism of their regulation, as previously described [23].

We have shown that the P1 promoter has a reduced activity compared to P2 or P1-P2 promoter constructs. Previous findings indicate that the P1 promoter is responsible for the transcription of 10–25% of *c-MYC*; however, P2 generates 75–90% of the transcripts [24]. Higher P2 activity in hepatic cells can also be explained by the presence of regulatory elements in its sequence, such as the TATA box, which has been described as the most appropriate to account for the high rate of inducible transcription [24,25]; moreover, the *c-MYC* P2 promoter contains two initiator elements [26], as well as other trans-activating factor binding sites [27], which include transcription factors such as YY1, Sp1, Sp3, AP-2, among others [24]. 

Isotretinoin is often used to treat severe acne and is considered a strong teratogen and trypanocide [28]. Isotretinoin has been studied for its epigenetic effects and has been shown to be a strong DNA demethylation agent [29,30]. On the other hand, thalidomide is a sedative-hypnotic and multiple myeloma medication [30], and is also considered a strong teratogenic drug [31]. The mechanisms of action of both isotretinoin and thalidomide remain poorly understood; however, the observed levels of *c-MYC* mRNA in this study indicate that the drugs may act as antagonists of each other. In fact, the same proteins are modified by both treatments according to the proteome profile results, suggesting that both drugs share the same pathways. For example, isotretinoin can modulate the expression of transforming growth factor-beta (TGF-β), and Leivo and colleagues demonstrated that oral isotretinoin (13-*cis*-retinoic acid) modified the expression of two distinct isoforms of TGF-β in suction blister fluid and serum in acne patients [32]. Moreover, *TGF-β1* gene expression was attenuated by all-trans retinoic acid, and even more so with isotretinoin treatment in Thy-GN rats [33]. Retinoids also reduce levels of TGF-β signaling superfamily member BMP4 by enhancing ubiquitin-mediated degradation of pSMAD1. The metabolite 9-*cis*-retinoic acid suppresses TGF-β-mediated induction of pro-fibrotic molecules in cultures of human mesangial cells, and this effect is mediated by the stimulation of hepatocyte growth factor [34]. Therefore, there may be crosstalk between the metabolite tretinoin or 9-*cis*-retinoic acid and TGF-β signaling, which ultimately has an effect on SMADs; there are at least five binding sites for SMADS along the *c-MYC* promoter [35]. Expression of transcription factor AP-2α (TFAP2A or AP-2) can be altered by retinoic acid exposure [36], AP-2 and c-MYC interact at the protein level and AP-2 overexpression decreases *c-MYC* mRNA expression [37,38]. Furthermore, there are at least two known binding sites for AP-2 on *c-MYC* promoter [39]. Therefore, isotretinoin has a number of pathways by which it could exert its effects on P1 or P2 *c-MYC* promoters.

On the other hand, thalidomide is an immunomodulatory drug and a strong teratogen that has been shown to interact with several transcription factors that can bind to P1 and/or P2 *c-MYC* promoter elements. For instance, it has been shown that thalidomide inhibits NF-κB though suppression of IκB activity (binding site in the P2 *c-MYC* promoter) [40]. Another transcription factor that thalidomide inhibits is Sp1 [41]. Sp1 and NF-κb have more than one binding site in the *c-MYC* promoter. Liu and colleagues reported that thalidomide is capable of inhibiting TGF-β1-mediated non-Smad ERK1/2 signaling pathways [42], while other studies demonstrated that thalidomide can exert its effects via inhibition of the TGF-β1 signaling pathway in an animal model of liver cirrhosis [43] and alveolar epithelial cells [44]. We have demonstrated that even though thalidomide does not have an effect mediated through P1 or P2, it still reduces *c-MYC* mRNA expression. As discussed, TGF-β is one of the pathways by which both isotretinoin and thalidomide could antagonise each other, leading to the contrary results of *c-MYC* mRNA expression.

The proteome profile assay showed that isotretinoin and thalidomide similarly regulate a set of proteins, oncoproteins and tumor suppressor proteins, which reinforces the idea that these two drugs modulate similar signaling pathways. Among the proteins that were similarly modified by each drug, the up-regulation of Decorin, DLL1, ERalpha, SNAIL, AXL and CEACAM-5, and the down-regulation of M-CSF are notable. In this context, Decorin, a small leucine-rich proteoglycan of the extracellular matrix, has been established as a tumor suppressor gene important in the prevention of liver cancer [45]. Another relevant finding was the down-regulation of macrophage colony-stimulating factor (M-CSF). Although M-CSF has been mainly implicated in the development of breast cancer, Akazawa and colleagues demonstrated a correlation between M-CSF inhibition and a decrease in liver tumor size in a mouse model [46]. In addition, we observe modifications of DDL1 and ERalpha proteins, both with reported oncogenic and tumor suppressor roles in cancer. The Delta-like ligand 1, DLL1, has been identified as a protein with dual roles in various types of cancer [47]. It is presumed that this dual capacity is dependent on the type of DLL1-binding ligand, which can activate or inactivate the Notch pathway. Likewise, estrogen receptor alpha (ERalpha) can promote or inhibit cancer progression, depending on the cancer type [48]. Although the proteome profile assay showed favorable results by modulating Decorin and M-CSF proteins, our analysis also demonstrates an up-regulation of the cancer-related proteins AXL [49], CEACAM-5 [50], and SNAIL [51] due to the effect of the drugs. It is important to consider that the proteome profile assay evaluates the expression of proteins implicated in different types of cancers, and some of the up-regulated oncoproteins have not been directly implicated in HCC. Also, the use of antibody-based arrays is an attractive approach for the identification of molecular targets involved in cancer; however, post-validation is essential to confirm the results. In this context, the validation assays by inmunoblot of M-CSF and SNAIL proteins did not show a correlation with the previously observed results of the protein array. These differences could be attributed to the molecular protein complexity of the sample and denaturation affecting immunoreactivity [52,53]. Complementary studies are necessary to determine the effect of isotretinoin and thalidomide on these cancer-related proteins.

In conclusion, isotretinoin and thalidomide down-regulate *c-MYC* mRNA expression and this is partially due to P1 or P2 promoter activity, which was shown to be active in HepG2 cells. Several pathways may be involved in the down-regulation of *c-MYC* mRNA. It should be considered that metabolites may play an essential role in the activation of different pathways. The main result of this work is that the repurposing of isotretinoin and thalidomide alone or in combination with other HCC-specific drugs, such as sorafenib, is a promising option for modulation of the expression of oncogenes and tumor suppressor genes for the treatment of HCC.

## 4. Materials and Methods

### 4.1. Chemicals

Isotretinoin (catalog #4759-48-2, purity ≥98%), thalidomide (catalog #50-35-1, Purity ≥ 98%), and dimethyl sulfoxide (DMSO, catalog D8418) were purchased from Sigma-Aldrich (St. Louis, MO., USA).

### 4.2. Cell Culture and Treatments

HepG2 cells, obtained from the American Tissue Culture Collection (ATCC HB-8065), were cultured in DMEM high glucose supplemented with 2.5 mM L-glutamine, 10% fetal bovine serum (Gibco, Carlsbad, CA, USA), 1 mM sodium pyruvate (Corning, Manassas, VA, USA), and 1% non-essential amino acids. Cultures were maintained at 37 °C in a humified atmosphere with 5% CO_2_. Cells were treated with 10 μM isotretinoin and/or 10 μg/mL (38.72 mM) thalidomide for 48 h, unless otherwise stated. Isotretinoin and thalidomide were diluted in culture medium from a DMSO stock solution. For all treatments DMSO concentration was less than 0.1%.

### 4.3. Viability Assays

Cell viability was determined using CellTiter-Blue Cell Viability Assay (Promega, Madison, WI, USA) according to the manufacturer’s instructions. HepG2 cells were seeded into 96-well plates at a density of 2 × 10^3^ cells per well. Every 24 h for a period of 96 h, fresh media with isotretinoin and thalidomide at indicated concentrations were added to each well. Prior to being assayed, pre-warmed fresh medium with CellTiter-Blue reagent was added to each well. Plates were gently shaken and incubated for 3 h at 37 °C with 5% CO_2_. Before measurements, the reaction was stopped and stabilized with 3% SDS. Drug sensitivity was determined by absorbance at wavelengths of 570 nm/600 nm using the 800 TS Absorbance Reader (BioTek Instruments, Winooski, VT, USA). Data were normalized to untreated cells as control.

### 4.4. RNA Extraction 

For total RNA extraction from cells, TRIzol reagent (Thermo Fisher Scientific, MA, USA) was used according to provider’s recommendations. Briefly, after washing cells with Phosphate Buffer Saline (PBS) 1 mL TRIzol was added to untreated and drug-treated cells, and incubated for 10 min at room temperature. Then, 200 µL chloroform was added and mixed by inversion for 15 s. Samples were then incubated for 2 min at room temperature and centrifuged at 12,000 *g* for 15 min at 4 °C. Supernatants were recovered and 500 µL isopropanol was added, samples were then vortexed and incubated for 10 min at room temperature. Microtubes with samples were centrifuged at 12,000 *g* for 10 min at 4 °C and supernatant was discarded. Subsequently, RNA pellet was washed with cold 70% ethanol, and samples were centrifugated at 7,500 *g* for 5 min at 4 °C. Finally, ethanol was discarded, samples were left to dry and resuspended in 30 µL nuclease-free water (Invitrogen). RNA integrity was visualized by electrophoresis on an 1% agarose gel stained with GelRed (Biotium, Hayward, CA, USA). The concentration and purity were determined at 260 and 280 nm on a Nanodrop 2000 equipment (Thermo Scientific, Boston, MA, USA). Samples with values >1.7 of absorbance on the 260/280 ratio were used for further experiments. 

### 4.5. Reverse Transcription and Quantitative PCR Assays

Complementary DNA (cDNA) synthesis was performed in two steps. Reverse transcriptase M-MLV (Invitrogen) was used as per the provider’s instructions for this purpose. For the first step, 2 µg RNA were mixed with 1 µL of random primers and 1 µL dNTPs (deoxynucleotide triphosphates) in a microtube and incubated for 5 min at 65 °C. For the second step, 4 µL first strand DNA buffer 5×, 2 µL DTT and 1 µL RNAse Out were added to the first tube and incubated for 2 min at 37 °C. Finally, 1 µL M-MLV reverse transcriptase was added to the reaction tube and incubated for 10 min at 25 °C, followed by incubation at 37 °C for 50 min and the inactivation of the enzyme for 15 min at 70 °C. cDNA functionality was determined by the amplification of the constitutive ribosomal 18S gen by end-point PCR using the primers Forward (5’-GTT ATT TCC AGC TCC AAT AGC GTA-3’) and Reverse (5’-GAA CTA CGA CGG TAT CTG ATC GTC-3’), and visualized in a 1% agarose gel stained with GelRed. Quantitative PCR (qPCR) assays were performed with TaqMan probes (Applied Biosystems, Carlsbad, CA, USA) for *c-MYC* gene (Hs00153408_m1) on a 7500 Fast Real Time PCR system (Applied Biosystems). The PCR reaction mix contained 10 µL TaqMan Universal Master Mix, 1 µL TaqMan probe/primers 20×, 2 µL cDNA and 7 µL nuclease-free water. The amplification was performed on standard conditions with an initial incubation at 50 °C for 2 min, followed by 95 °C for 10 min and 40 cycles at 95 °C for 15 s and 60 °C for 1 min. A dynamic amplification range with serial dilutions of DNA from 1:16 to 1:1,024 was performed, and 1:128 dilution was chosen as the most appropriate. All qPCR experiments were performed in triplicate. *GAPDH* gene (Cat. Hs99999905_m1) was used as endogenous control and its amplification was run in parallel. Results were obtained and processed using the 2^(-∆∆CT) formula [54].

### 4.6. Constructs, Transient Transfections and Promoter Activity Assays 

pGL3-Myc-P1 construct (containing promoter P1 from -1,375 to −38, with respect to promoter P2) was obtained by subcloning an amplified fragment of 1,338 bp from human genomic DNA with primers MycP1-F (5’-AGA GGC TAG CGG GAA AAG AGG ACC TGG AAA G-3′) and MycP1-R (5′-AGA GGC TAG CGA TCC CTC CCT CCG TTC TT-3′), where the underlined sequence is the *Nhe*I recognition site, into the pGL3Basic vector (Promega, Madison, WI. USA) cut with *Nhe*I enzyme. Sequence fidelity and orientation was confirmed by automated sequencing using primer GLprimer2 (5’-CTT TAT GTT TTT GGC GTC TTC CA-3’) and the BigDye terminator v3.1 cycle sequencing kit (Applied Biosystems, CA) in a 3130*xl* genetic analyzer (Applied Biosystems). pGL3-Myc-P2 (from −92 to +50) and pGL3-Myc-P1-P2 (from −2,450 to +50) constructs were a generous gift from Dr. Michael Cole (Norris Cotton Cancer Center, New Hampshire, USA). The c-MYC promoter constructs fused with the luciferase gene were transfected in HepG2 cells using Lipofectamine 2000 reagent (Invitrogen, CA., USA) as previously described [55]. Briefly, 1 μg pGL3-Myc-P1, pGL3-Myc-P2 or pGL3-Myc-P1-P2 and 100 ng phRL-CMV plasmids (as a control for normalizing transfection efficiency) were incubated with 250 μL DMEM without serum for 5 min. In another microtube, the plasmids were mixed with 10 μL Lipofectamine 2000, previously diluted in 250 μL serum-free DMEM. Tubes were incubated for 20 min at room temperature, and DNA-Lipofectamine complexes were added to 2 × 10^5^ cells. For each assay, pGL3 Basic Vector and pGL3 Control Vector (Promega) were transfected in parallel as negative and positive controls, respectively. The medium was replaced with DMEM supplemented with 10% fetal bovine serum after 5 h of incubation. Twenty-four hours after transfection, the cells were exposed either to 10 μM isotretinoin or 10 μg/mL thalidomide. After 48 h of incubation, cells were homogenized with Passive Lysis Buffer (Promega) for 15 min on an oscillatory shaker at room temperature. Firefly and Renilla luciferase activity were measured with the Dual-Luciferase Assay System (Promega) in a Modulus Luminometer (Turner BioSystems, Sunnyvale, CA, USA). the Renilla luciferase activity normalized to Firefly luciferase activity, and DMSO-treated cells were set as 100%. Blanks were analyzed by conducting luciferase activity assays in untransfected cells. 

### 4.7. Proteome Array and Densitometric Analysis 

Relative protein expression levels of 84 human cancer-related proteins were analyzed by the Proteome Profiler Human XL Oncology Array Kit (Cat# ARY026; R&D systems, MN) according to the manufacturer’s recommendations. Briefly, membranes were placed in separate wells of the 4-well Multi-Dish with 2 mL of Array Buffer 6 for 1 h. Then, 200 μg total protein combined with Array Buffer 4 and Array Buffer 6 were added to the membranes and incubated overnight at 4 °C. On the next day, membranes were washed 3 times for 10 min with Washing Buffer solution and incubated with the Detection Antibody cocktail for 1 h. Subsequently, each of the membranes was washed 3 times and then incubated in 2 mL streptavidin-HRP for 30 min. The membranes were washed and covered with 1 mL Chemi Reagent Mix for 1 min and then exposed to an X-ray film in the dark at room temperature. For densitometric analysis, the Fiji software v1.51 (at https://imagej.net/software/fiji/) was used. Values were processed by WebMeV (Multiple Experiment Viewer at https://webmev.tm4.org/#/about) for visualization and stratification of data.

### 4.8. Western Blotting

Cells were scrapped from cell culture dishes and transferred to 1.5 mL microtubes with PBS and protease inhibitor cocktail (Roche, Mannheim, Germany). Cells were lysed by thermal shock in five cycles of cold (dry ice) and heat (37 °C) for 1 min each cycle. The samples were centrifuged at 5,000 rpm for 5 min at 4 °C, supernatant was recovered and protein concentration was determined with Bradford assay method. Then, 30 μg protein extracts were resolved by electrophoresis using a 10% SDS-polyacrylamide gel under denaturing conditions. Proteins were electrotransferred to a polyvinylidene fluoride membranes (Millipore, Bedford, MA, USA) and then blocked with 5% skim milk (Svelty low fat; Nestlé) in TBST buffer for 1 h. Membranes were incubated overnight at 4 °C with the primary antibody diluted in blocking solution. The membranes were washed with TBST buffer and incubated with the corresponding secondary antibody for 1 h at room temperature. Immunoblots were detected by chemiluminescence using the ChemiDoc XRS + System (Bio-Rad, Hercules, CA, USA). The following primary and secondary antibodies were used: anti-c-MYC (1:1,000 dilution) (Cat. 3C117, Santa Cruz Biotechnology), anti-Actin (1:2,000 dilution) (Cat. ab8227, Abcam,), anti-M-CSF (1:10,000 dilution) (Cat. 143650, USBiological), anti-SNAIL (1:4,000 dilution) (Cat. BS91262, Bioworld Technology), anti-p53 (1:1,000 dilution) (Cat. Ab26, Abcam), peroxidase-labeled anti-mouse (1:20,000 dilution) (Cat. NIF825; GE Healthcare Life Science), and mouse anti-rabbit IgG-HRP (1:5,000 dilution) (Cat. sc-2357, Santa Cruz Biotechnology). Protein expression levels were quantified with the Image Studio Lite software (LI-COR Biosciences, NE, USA) and normalized with actin values. 

### 4.9. Statistical Analysis 

Two-way ANOVA with Dunnett multiple comparison tests were used for viability assays. For mRNA expression assays, luciferase activity, and inmunoblots data were analyzed by non-parametric Mann–Whitney U test with the SPSS v2.0 software (Armonk, New York, NY, USA) and GraphPad Prism 6 (San Diego, CA, USA).

## Figures and Tables

**Figure 1 molecules-26-05742-f001:**
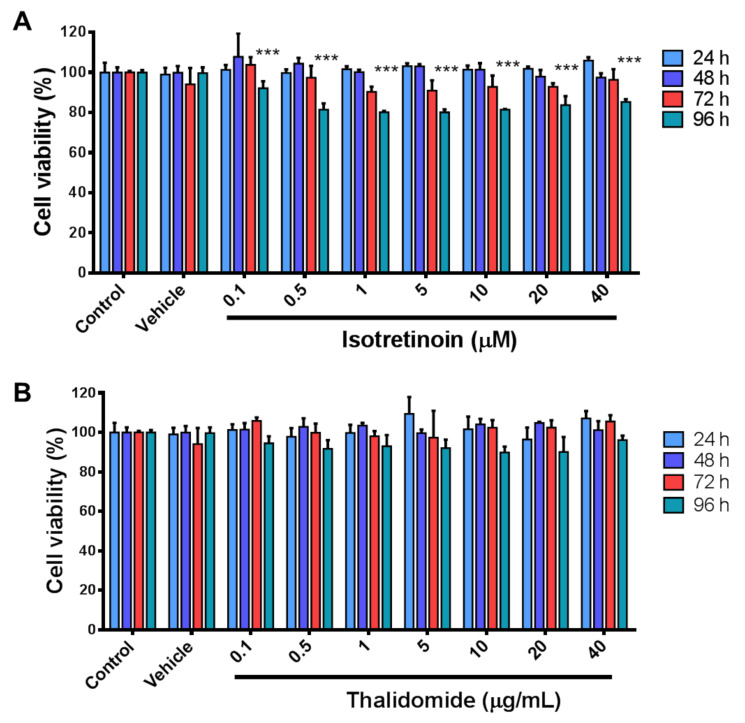
Cell viability assays in HepG2 cells in response to isotretinoin and thalidomide. Cells were exposed to different concentrations of isotretinoin (**Panel A**) and thalidomide (**Panel B**) for up to 96 h. Untreated cells (control) and Dimethyl sulfoxide (DMSO)-exposed cells (vehicle) were used as references. DMSO concentration was less than 0.1%. Values are expressed as mean ± SD from three independent experiments. *** *p* < 0.001.

**Figure 2 molecules-26-05742-f002:**
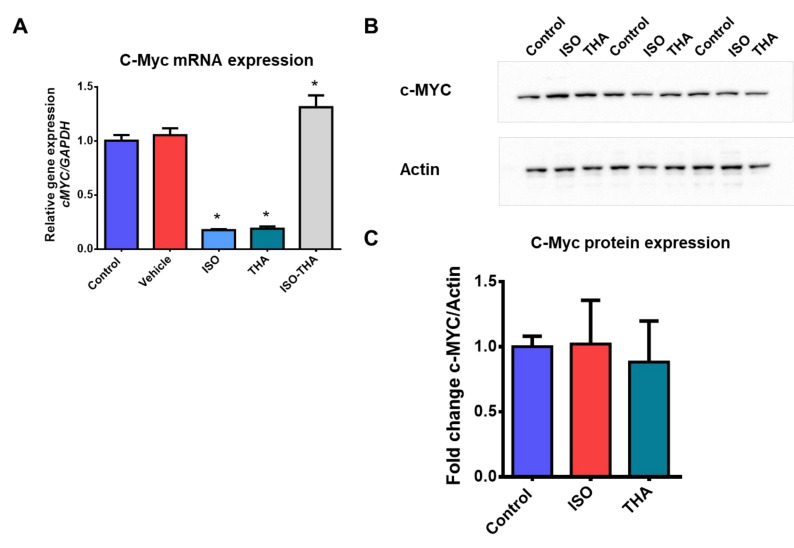
c-MYC expression in HepG2 cells exposed to epigenetic drugs. **Panel A**, cells were exposed to 10 µM isotretinoin (ISO) and/or 10 µg/mL thalidomide (THA) for 48 h. RT-qPCR analysis was performed for the *c-MYC* gene normalized to *GAPDH* expression. Cells treated with DMSO (vehicle) were used as references. DMSO concentration was less than 0.1%. **Panel B**, protein extracts from cells exposed to 10 µM isotretinoin (ISO) and/or 10 µg/mL thalidomide (THA) for 48 h were obtained and western blot assays were performed using anti-c-MYC and anti-actin antibodies. Assays were performed by triplicate, and controls DMSO-treated were included for comparisons. **Panel C**, graphical representation of normalized c-MYC protein content in cells exposed to isotretinoin or thalidomide. Values are expressed as mean ± SD from six independent experiments. * *p* < 0.05.

**Figure 3 molecules-26-05742-f003:**
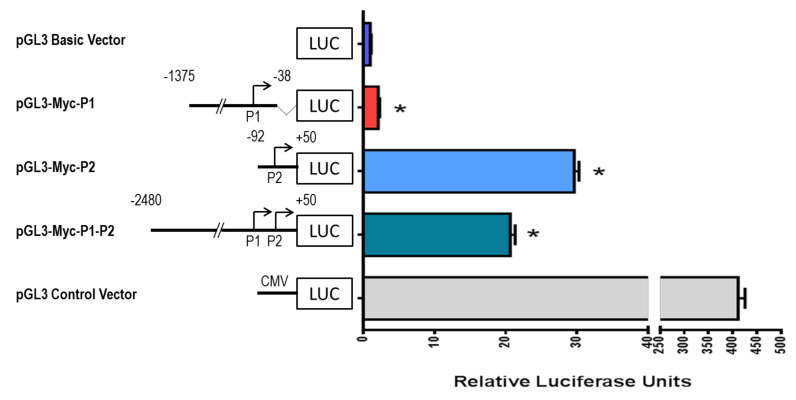
Promoter activity of *c-MYC* in HepG2 cells. Fragments bearing P1 and/or P2 c-MYC promoters were subcloned into the pGL3 Basic Vector, transfected into HepG2 cells using cationic liposomes, and after 48 h, cells were lysed to perform luciferase activity assays. pGL3 Control Vector was used as a positive control, and pGL3 Basic Vector was used as a negative control. Values are expressed as mean ± SD from three independent experiments. * *p* < 0.05.

**Figure 4 molecules-26-05742-f004:**
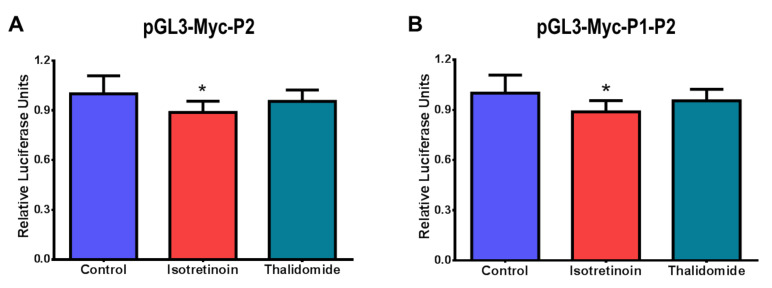
Promoter activity of c-MYC in HepG2 cells exposed to isotretinoin and thalidomide. Cells were transfected with pGL3-Myc-P2 (**Panel A**) or pGL3-Myc-P1-P2 (**Panel B**) and exposed to 10 µM isotretinoin or 10 µg/mL thalidomide for 48 h. Cells treated with DMSO (Control) were used as references. DMSO concentration was less than 0.1%. Cells were lysed to perform luciferase activity assays. Values are expressed as mean ± SD from three independent experiments. * *p* < 0.05.

**Figure 5 molecules-26-05742-f005:**
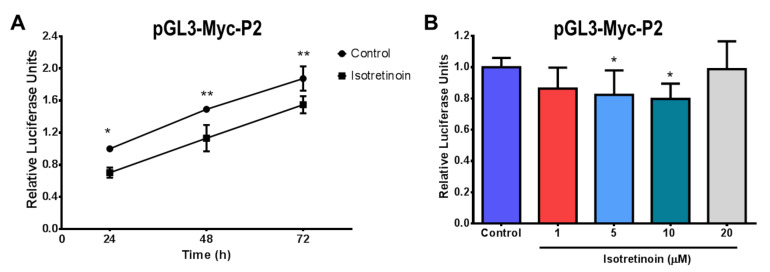
P2 promoter activity of *c-MYC* in HepG2 cells exposed to isotretinoin. **Panel A**, cells were transfected with pGL3-MYC-P2 and exposed to 10 µM isotretinoin for 24, 48, and 72 h. **Panel B**, cells were transfected with pGL3-MYC-P2 and exposed to 1, 5, 10, and 20 µM isotretinoin for 24 h. Cells treated with DMSO (Control) were used as references. DMSO concentration was less than 0.1%. Cells were lysed to perform luciferase activity assays. Values are expressed as mean ± SD from three independent experiments. * *p* < 0.05, ** *p* < 0.01.

**Figure 6 molecules-26-05742-f006:**
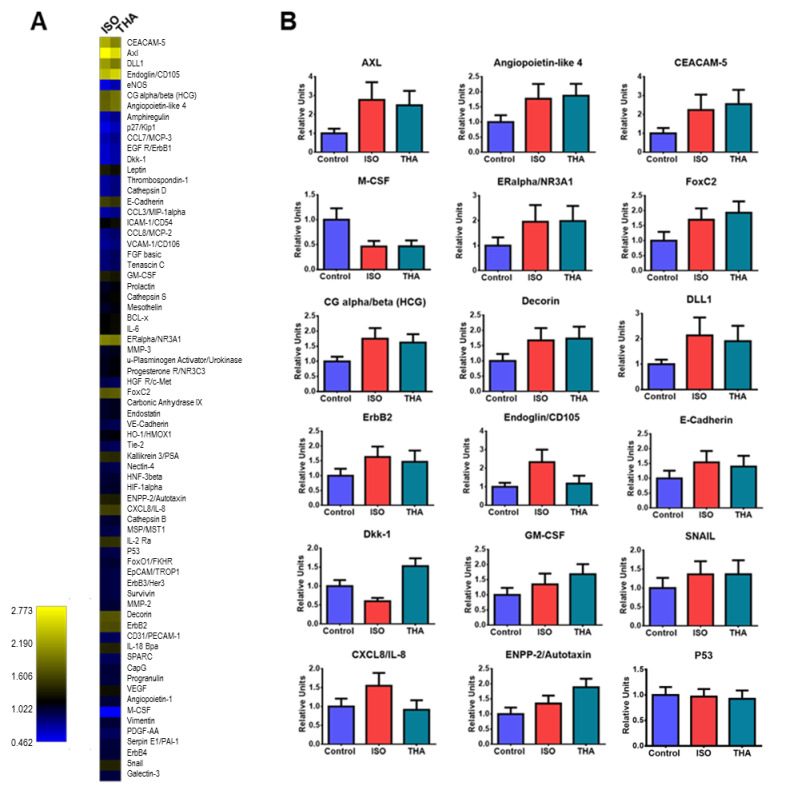
Expression of human proteins related to cancer pathogenesis in HepG2 cells exposed to isotretinoin and thalidomide. Cells were exposed to 10 µM isotretinoin (ISO), 10 µg/mL thalidomide (THA) or DMSO (control). Cell extracts were obtained and processed as described in Materials and Methods. **Panel A**, Heat map to visualize relevant protein expression changes due to isotretinoin or thalidomide treatment. Data were normalized using the results from untreated cells. **Panel B**, graphical representation of protein levels calculated by densitometry. Analysis of protein expression was performed by duplicate.

**Figure 7 molecules-26-05742-f007:**
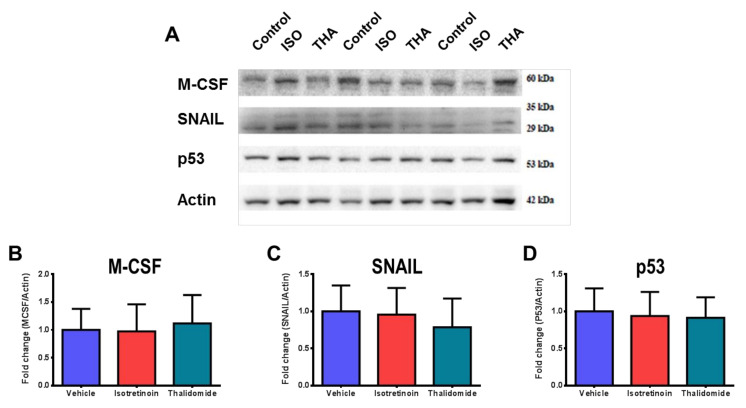
Inmunoblot analysis of M-CSF, SNAIL, and P53 in HepG2 cells exposed to isotretinoin and thalidomide. Cells were exposed to 10 µM isotretinoin (ISO) or 10 µg/mL thalidomide (THA) for 48 h. **Panel A**, protein extracts were obtained from untreated and treated cells, and Sodium dodecyl-sulfate polyacrylamide gel electrophoresis (SDS-PAGE) using specific antibodies was performed. DMSO-exposed cells (control) were used as references. DMSO concentration was less than 0.1%. Graphical representation of densitometric analyses of M-CSF (**Panel B**), SNAIL (**Panel C**), and p53 (**Panel D**). Values are expressed as mean ± SD from three independent experiments.

## Data Availability

The data presented in this study are available on request from the corresponding author.

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
