# Peer review of "Isotretinoin and Thalidomide Down-Regulate c-MYC Gene Expression and Modify Proteins Associated with Cancer in Hepatic Cells"

_molecules, 2021, doi:10.3390/molecules26195742_

Round 1

Reviewer 1 Report

In this manuscript, the authors performed a study on Isotretinoin and thalidomide down-regulate c-MYC gene expression and modify proteins associated with cancer in hepatic cells . The authors hope to provide a new therapeutic strategy for hepatocellular carcinoma. The authors demonstrated that Isotretinoin and thalidomide can downregulate mRNA levels of c-myc, but not significantly at the protein level. The authors attribute the phenomenon to protein half-life. However, it is known that proteins are the most important functional structures in cells. If the effect of the drug cannot be detected at the protein level, it is difficult to show that this is a promising therapeutic option. The authors need to explore the reasons in more depth. In addition, few studies have reported the use of Isotretinoin and thalidomide, especially Isotretinoin, in hepatocellular carcinoma. From the data given by the authors, the in vitro activity of both compounds is also not good. The authors need to explain the idea of the subject design. In summary, this study is not a complete study at this time and I recommend rejection of the manuscript.

Author Response

Comment:

this manuscript, the authors performed a study on Isotretinoin and thalidomide down-regulate c-MYC gene expression and modify proteins associated with cancer in hepatic cells . The authors hope to provide a new therapeutic strategy for hepatocellular carcinoma. The authors demonstrated that Isotretinoin and thalidomide can downregulate mRNA levels of c-myc, but not significantly at the protein level. The authors attribute the phenomenon to protein half-life. However, it is known that proteins are the most important functional structures in cells. If the effect of the drug cannot be detected at the protein level, it is difficult to show that this is a promising therapeutic option. The authors need to explore the reasons in more depth.

Response: We agree with the comment that proteins are the most important molecules in functional processes than mRNAs. Previous results from our group and others [1-5], indicate that mRNA expression in human/murine tissue is an accurate marker of protein presence. Nevertheless, attempts to correlate protein abundance with mRNA expression levels have had variable success. Regulation of gene expression is tightly controlled at the levels upstream of translation and not always correlates directly [6, 7]. Epigenetic regulation is well documented in processes such has metabolic diseases, obesity, diabetes mellitus, aging, or even other pathologies, like cancer [8]. There is growing evidence supporting the role of drugs on epigenetic modifications to improve, reduce or exacerbate cancer, and more mechanisms are being elucidated, enabling our understanding on how epigenetics could be used for novel strategies in prevention, treatment or to shift cancer. Although our first approach was on c-Myc expression, we have revealed that other proteins were also modified by both compounds with similar patterns,,as DKK1, which has been linjed to c-Myc regulation [9]. Then, it is possible to complement our results with additional experiments in the future, as the possibility to use these molecules in combination with sorafenib, the prescribed drug for HCC, to improve the outcome. However, many preclinical and clinical studies are necessary to perform to conclude about the promising therapeutic option of these molecules.  

Comment: In addition, few studies have reported the use of Isotretinoin and thalidomide, especially Isotretinoin, in hepatocellular carcinoma. From the data given by the authors, the in vitro activity of both compounds is also not good. The authors need to explain the idea of the subject design. In summary, this study is not a complete study at this time and I recommend rejection of the manuscript.

Response: We thank the reviewer comment. Our main idea was to identify which drugs are able to inhibit c-Myc expression in HCC. However, as we showed, c-Myc mRNA is down-regulated by these two drugs, but not c-Myc protein. On the other hand, as we commented before, proteome profile reveals oncogene and suppressor tumor genes as targets of these compounds. Implications of this phenomenon should be explored in further experiments. We understand his/her reviewer concern, but we think that this first finding on differential regulation of c-Myc could contribute to identify other mechanisms/drugs that could impact, or not, at protein level.

References

  1. Becerril-Esquivel, C.; Penuelas-Urquides, K.; Blancas-Sanchez, E.; Zapata-Benavides, P.; Silva-Ramirez, B.; Chavez-Reyes, A.; Castorena-Torres, F.; Cisneros, B.; Bermudez de Leon, M., The polyaromatic hydrocarbon beta-naphthoflavone alters binding of YY1, Sp1, and Sp3 transcription factors to the Dp71 promoter in hepatic cells. Mol Med Rep 2018, 17, (4), 6150-6155.
  2. Cordero-Reyes, A. M.; Youker, K.; Estep, J. D.; Torre-Amione, G.; Nagueh, S. F., Molecular and cellular correlates of cardiac function in end-stage DCM: a study using speckle tracking echocardiography. JACC Cardiovasc Imaging 2014, 7, (5), 441-52.
  3. Gupte, A. A.; Hamilton, D. J.; Cordero-Reyes, A. M.; Youker, K. A.; Yin, Z.; Estep, J. D.; Stevens, R. D.; Wenner, B.; Ilkayeva, O.; Loebe, M.; Peterson, L. E.; Lyon, C. J.; Wong, S. T.; Newgard, C. B.; Torre-Amione, G.; Taegtmeyer, H.; Hsueh, W. A., Mechanical unloading promotes myocardial energy recovery in human heart failure. Circ Cardiovasc Genet 2014, 7, (3), 266-76.
  4. Lara-Chacon, B.; de Leon, M. B.; Leocadio, D.; Gomez, P.; Fuentes-Mera, L.; Martinez-Vieyra, I.; Ortega, A.; Jans, D. A.; Cisneros, B., Characterization of an Importin alpha/beta-recognized nuclear localization signal in beta-dystroglycan. J Cell Biochem 2010, 110, (3), 706-17.
  5. Aguirre-Vazquez, A.; Salazar-Olivo, L. A.; Flores-Ponce, X.; Arriaga-Guerrero, A. L.; Garza-Rodriguez, D.; Camacho-Moll, M. E.; Velasco, I.; Castorena-Torres, F.; Dadheech, N.; Bermudez de Leon, M., 5-Aza-2'-Deoxycytidine and Valproic Acid in Combination with CHIR99021 and A83-01 Induce Pluripotency Genes Expression in Human Adult Somatic Cells. Molecules 2021, 26, (7).
  6. Plotkin, J. B., Transcriptional regulation is only half the story. Mol Syst Biol 2010, 6, 406.
  7. Vogel, C.; Abreu Rde, S.; Ko, D.; Le, S. Y.; Shapiro, B. A.; Burns, S. C.; Sandhu, D.; Boutz, D. R.; Marcotte, E. M.; Penalva, L. O., Sequence signatures and mRNA concentration can explain two-thirds of protein abundance variation in a human cell line. Mol Syst Biol 2010, 6, 400.
  8. Whayne, T. F., Epigenetics in the development, modification, and prevention of cardiovascular disease. Mol Biol Rep 2015, 42, (4), 765-76.
  9. Liu, W.; Fu, X.; Li, R., CNN1 regulates the DKK1/Wnt/beta-catenin/c-myc signaling pathway by activating TIMP2 to inhibit the invasion, migration and EMT of lung squamous cell carcinoma cells. Exp Ther Med 2021, 22, (2), 855.

Reviewer 2 Report

In this manuscript, Patricia Nefertari Ramirez-Flores et al showed that  These findings are interesting. The manuscript could be further strengthened with some of the correction denoted below.

  1. In figure 1, data seems that cell viability increases over time when processing by concentration both Isotretinoin and Thalidomide. Authors need to more explain about this.
  2. In figure 2B, authors need to indicate more detail information on the data.
  3. In figure 5A, please put on the 0 h data in the graph.
  4. In figure 6, please show the better quality of data.
  5. The connection is weak to explain about p53 in figure 7. Please explain more about why authors conducted the experiment about p53.
  6.  There are many places that incorrectly or inaccurately write down the manuscript. Authors need to pay close attention to proper labeling of this manuscript as well.

Author Response

Comment: In this manuscript, Patricia Nefertari Ramirez-Flores et al showed that  These findings are interesting. The manuscript could be further strengthened with some of the correction denoted below.

In figure 1, data seems that cell viability increases over time when processing by concentration both Isotretinoin and Thalidomide. Authors need to more explain about this.

Response: We thank the reviewer observation, however the slight increase observed in Fig. 1A and 1B are not significant. We have revised the literature, and no information is available if both compounds promote cell proliferation. Then, we conclude that there is a non-significant increase in the cell viability assays.

Comment: In figure 2B, authors need to indicate more detail information on the data.

Response: We incorporated more detailed information for Fig. 2B in Results section and figure legend.

Comment: In figure 5A, please put on the 0 h data in the graph.

Response: Done.

Comment: In figure 6, please show the better quality of data.

Response: We improved the quality of data in Figure 6.

Comment: The connection is weak to explain about p53 in figure 7. Please explain more about why authors conducted the experiment about p53.

Response: We decided to include p53 analysis because is a key factor in cancer development. Moreover, p53 is included in the proteome array showed in Fig. 6A; then, as part of validation of the array, we performed the p53 detection using western botting.

Comment: There are many places that incorrectly or inaccurately write down the manuscript. Authors need to pay close attention to proper labeling of this manuscript as well.

Response: We thank for the reviewer suggestion. We have reviewed all document to correct the information and proper labeling.

Round 2

Reviewer 1 Report

the manuscript can be accept now.